# The views of UK Chinese individuals towards living and deceased-donor kidney transplantation: A qualitative interview study

**Matthew Beresford** [1,2]*, **Katie Wong**[1,3], **Mohammed Al-Talib**[1,4], **Pippa K. Bailey**[1,5]

1 Population Health Sciences, Bristol Medical School, Bristol, United Kingdom, 2 Department of Nephrology, Gloucestershire Hospitals NHS Foundation Trust, Gloucester, United Kingdom, 3 Department of Renal Medicine, National Registry of Rare Kidney Diseases, University College London, London, United Kingdom, 4 Systems Immunity University Research Institute/Division of Infection and Immunity, School of Medicine, Cardiff University, Cardiff, United Kingdom, 5 Department of Nephrology and Transplantation, North Bristol NHS Trust, Bristol, United Kingdom

* matthew.beresford2@nhs.net

## Abstract

The views of UK Chinese people towards transplantation and organ donation are not known. It is not known whether the perspectives of Chinese people living in the UK differ from those of Chinese people living elsewhere in the world, or whether perspectives of UK Chinese people vary according to time spent living in the UK. This qualitative interview study aimed to investigate the views of UK Chinese individuals towards kidney donation and transplantation. It formed part of a convergent parallel mixed-methods programme of research alongside a quantitative registry study which found that UK Chinese individuals experience poorer access to living-donor kidney transplantation compared to UK White individuals. We conducted in-depth semi-structured interviews with sixteen participants across three UK cities. Participants were permanently resident in the UK and self-identified as UK Chinese. Interviews were conducted between 9th April 2020 and 16th July 2020. Interviews were transcribed verbatim, coded using NVivo software, and analysed using inductive thematic analysis. Three main themes and seven sub-themes were identified: (1) Importance of kinship: biological and social (i) Familism, ii) Relationship hierarchy, iii) Matching; (2) Donor sacrifice (i) Negative impact on donors, ii) Bodily integrity after death; and (3) Patient as information gatekeeper (i) Culture of silence, ii) A perceived need for education and engagement. This study provides insights that may offer some explanation for reduced rates of living-donor transplantation amongst UK Chinese individuals. Further research is required to investigate observational research findings not explained here, and to develop effective strategies to improve treatment access for UK Chinese individuals with kidney disease.

**Data availability statement:** Anonymised transcripts of the interviews have been uploaded to the University of Bristol's Research Data Repository: https://data.bris.ac.uk/data/; (DOI 10.5523/bris.2cgaw1go9gtyo1z163m822wscs). One participant did not consent to data sharing, so their interview transcript has not been uploaded. Audio files of the recorded interviews are not suitable for sharing as they carry a high risk of allowing the research participant to be identified, and the content of interviews includes sensitive information. Individuals who wish to access the dataset can contact the researchers directly or search the University of Bristol's Research Data Repository. The dataset name is 'Interviews with UK Chinese Individuals' and the creator is Pippa Bailey. Although the qualitative transcripts have been anonymised, as personal and sensitive issues have been discussed we cannot rule out the risk of identification, and therefore access to these transcripts is controlled. Individual researchers will need to request access to the controlled data through the University of Bristol via the Data Access Committee (DAC) for approval, before data can be shared after their host institution has signed a Data Access Agreement. The procedure for accessing data can be found here: https://www.bristol.ac.uk/staff/researchers/data/accessing-research-data/.

**Funding:** The author(s) received no specific funding for this work.

**Competing interests:** The authors have declared that no competing interests exist.

## Introduction

The 2013 UK renal research strategy called for research to better understand ethnic variation in rates of kidney disease and access to treatments in the UK [1]. The UK 2021 census reported that 0.8% of the UK population (n = 502,216) were of Chinese ethnicity: a number that increased by 27% between 2011 and 2021 [2,3]. According to the UK Renal Registry (UKRR), 0.75% of people with kidney failure living in the UK are of Chinese heritage (2021, UKRR personal communication). This differs from other UK ethnic minorities who are over-represented in the UK kidney failure population.

In our quantitative analysis of UKRR data, we found that UK Chinese people with kidney disease are less likely than UK White patients to receive a living-donor kidney transplant (LDKT) [4]. Similar ethnic disparities in access to LDKT have been observed amongst first time kidney transplant candidates in the US, with evidence of increasing disparity between Asian (including Chinese Americans) patients and those of White ethnicity in 2014 compared to 1999 [5]. In our UKRR study, we also found that UK Chinese men have lower odds of accessing pre-emptive waitlisting and transplantation compared to White men: the same was not found for UK Chinese women compared to UK White women. The reasons for these disparities are not understood. Before the move to opt-out deceased-donor registration in the England and Wales, UK Chinese individuals were under-represented on the organ donor register (0.3% of registrants compared to 0.7% of the population in England and Wales). However, since the introduction of the opt-out law, representation of the UK-Chinese population on the organ donor register has now increased (0.9% of active donor registrants).

The views of UK Chinese people towards transplantation and organ donation are not known. It is uncertain whether the perspectives of Chinese people living in the UK differ from those of Chinese people living elsewhere in the world, or whether perspectives of UK Chinese people vary according to time spent living in the UK. A retrospective cohort study from The Netherlands reported that the more time an individual had spent in the Netherlands, the more their attitudes towards kidney donation converged with those of native individuals [6]. In addition, previous survey-based research amongst migrant populations originating from the same country but emigrating to either USA or Spain, has shown that attitudes towards organ donation differ depending on host country [7].

In this study we aimed to investigate and understand the attitudes of UK Chinese individuals towards kidney donation and transplantation. We undertook this study in parallel to the quantitative registry analysis as part of a convergent parallel mixed-methods programme of research [8], aiming to investigate transplantation and donation in the UK Chinese population. We investigated the views of UK Chinese people born in the UK and those born in China or outside the UK. To our knowledge it is the first investigation of attitudes to transplantation and donation in the UK Chinese population.

## Materials and methods

### Study design and recruitment of participants

In this qualitative interview study, we undertook in-depth semi-structured interviews with individuals aged 18 and over who self-identified as UK Chinese. UK Chinese

included people who self-identified as being of Chinese heritage who were currently resident in the UK. It included people born in the UK or outside the UK, having subsequently emigrated.

Participants were recruited via a combination of convenience and snowball sampling between 1st October 2019 and 16th July 2020 [9]. We contacted the chairs of 12 UK Chinese Associations via phone and email with details of the study, asking them to identify potential participants and to distribute information to their membership bases. Potential participants were contacted via letter or email and provided with the option of receiving further information about the study. Individuals were subsequently sent a patient information leaflet if agreeable. Interviewees provided details of further eligible individuals who were then invited to participate in the same way.

## Data collection

In-depth semi-structured interviews were conducted between 9th April 2020 and 16th July 2020. Interviews as opposed to focus groups were chosen for data collection to allow a more detailed exploration of sensitive topic areas and to allow participants to pick a time for participation that best suited them. Interviews were conducted over the telephone due to the COVID-19 pandemic. All participants provided informed verbal consent to participation which was audio-recorded and transcribed. Interviews were conducted by one member of the study team (KW). KW is a hospital nephrologist and research fellow with formal training in qualitative research methods. She identifies as UK Chinese and knew two participants as acquaintances prior to the study. Matters relating to the research study had not been discussed between KW and these acquaintances prior to the interviews taking place.

A flexible topic guide was developed by the study team (see S1 File). This served as a broad guide of topics to discuss: the questions listed were not asked verbatim and evolved as interviews progressed. Participants were asked about their understanding of kidneys in health, kidney disease, and kidney replacement therapies including transplantation. They were asked about the personal factors that would influence their willingness to donate a kidney in life or after death, their willingness to receive a kidney transplant, and their perception of Chinese attitudes to transplantation more broadly. Particular attention was paid to the nature of relationships between potential donors and recipients, and the impact of Chinese cultural or religious practices on organ donation.

The following participant demographics were collected at the time of interview: age, self-reported gender, place of birth, educational level, and year of emigration to the UK (if applicable). Information on non-participation rates and reasons for non-participation was not collected.

Interviews were undertaken in English but participants were given the option of using a Cantonese translator. The veracity of translations was confirmed by KW, who speaks conversational Cantonese. When interpreters were used, only the interpreter's speech in English was transcribed; and analysis thereafter was of English transcriptions only. Interviews were digitally audio-recorded, transcribed verbatim, anonymised, and transcripts uploaded to NVivo software for analysis.

## Data analysis

We undertook an inductive thematic analysis as described by Braun and Clarke [10]. The research was informed by a critical realist position, which considers an individual account as constructed, but also accepts it as a description of events and personal experiences that have some basis in reality [11]. All transcripts were coded by MB and a subset of interviews were independently coded by PKB and KW. MB is a clinical academic research fellow with formal training in qualitative methods, and received guidance from PKB, an experienced qualitative researcher. Transcripts were read at least twice to gain familiarisation with the data. Following familiarisation the entire dataset was coded: coding was inductive, and data driven. Initial codes were generated by assigning descriptive labels to interesting features of the data and sections of text. Codes were then collated into potential themes based on shared properties and clusters of meaning within the dataset, keeping the research objectives in mind.

This study was undertaken in parallel with a quantitative analysis as part of convergent parallel mixed-methods programme of work [8]. The report has been written in line with COREQ reporting guidelines (see S2 File) [12]. Ethical approval to undertake this study was obtained from the University of Bristol Faculty of Health Science Research Ethics Committee (FREC) prior to study commencement.

## Results

Sixteen individuals participated in fifteen interviews. Two interviews were undertaken with the facilitation of a Cantonese translator. Two participants (husband and wife) wished to be interviewed together. This interview was conducted on speakerphone with questions asked once, with the interviewees able to provide individual responses or discuss and answer collectively. Interview lengths ranged from 39 to 78 minutes. Participant characteristics are presented in Table 1.

Our diverse sample included individuals who had experience of a friend or family member living with kidney disease and individuals who had not been personally exposed to kidney disease or its treatments. No participants reported living personally with kidney disease. We did not identify a clear relationship between the degree of familiarity with the subject matter and the themes that emerged: for example, those who had some awareness of a person's lived experience did not appear more supportive of organ donation and transplantation than those without this experience.

Three major themes were identified with seven subthemes (Fig 1):

Themes and illustrative quotes are presented in Table 2. We have highlighted where themes appeared to specifically relate to receiving or donating an organ or partaking in living or deceased donation. However, views and beliefs about organ donation and transplantation did not always apply to a specific 'mode' of transplantation and were inextricably linked or related to higher level beliefs.

**Table 1. Participant characteristics.**

| Participant number | City | Age range | Gender (self-reported) | Place of birth | Educational level | Year of emigration to UK |
|---|---|---|---|---|---|---|
| 1[1] | 1 | 18–25 | Woman | China | Postgraduate | 2019 |
| 2[1] | 1 | 26–35 | Man | China | Postgraduate | 2000 |
| 3 | 1 | 46–55 | Woman | China | Undergraduate | 2001 |
| 4 | 2 | 56–65 | Woman | Hong Kong | Undergraduate | 1967 |
| 5 | 2 | 26–35 | Woman | UK | Postgraduate | n/a |
| 6 | 2 | 46–55 | Woman | China | Secondary | Not disclosed |
| 7 | 2 | 26–35 | Man | UK | Postgraduate | n/a |
| 8 | 3 | 56–65 | Man | Hong Kong | Vocational | 1995 |
| 9 | 3 | 46–55 | Man | Hong Kong | Secondary | 1999 |
| 10 | 3 | 76–85 | Woman | Hong Kong | Undergraduate | 1958 |
| 11 | 3 | 76–85 | Woman | Hong Kong | Secondary | 1978 |
| 12 | 3 | 66–75 | Woman | Hong Kong | Vocational | 1972 |
| 13 | 3 | 46–55 | Woman | Mauritius | Secondary | 2014 |
| 14 | 3 | Not disclosed | Not disclosed | Hong Kong | Postgraduate | 2003 |
| 15 | 3 | 76–85 | Woman | India | Undergraduate | 1965 |
| 16 | 3 | 66–75 | Man | Hong Kong | Secondary | 1993 |

[1]These two participants were interviewed together.

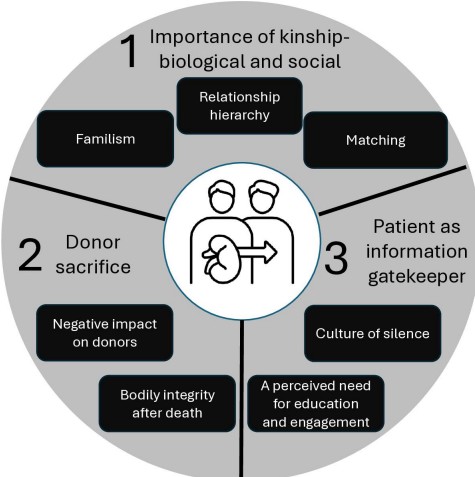

**Fig 1. Schematic depicting key themes and sub-themes.** 1. Importance of kinship: biological and social. a. Familism, b. Relationship hierarchy, c. Matching; 2. Donor sacrifice. a. Negative impact on donors, b. Bodily integrity after death; 3. Patient as information gatekeeper. a. Culture of silence, b. A perceived need for education and engagement.

**Table 2. Themes and illustrative quotes.**

| Theme | Subtheme | Quote |
|---|---|---|
| Importance of kinship: biological and social | Familism | "We all love our own family members. If they do need it I'm more than happy to donate because we can only live on one kidney." *Participant 11* |
| | | "Yeah, and also of course Chinese people I think, culture wise, I think Chinese people their close family, they're very close in culture, I mean in terms of family members, so I'm sure if you love someone in the family, if somebody have this problem, yeah, they will very easily say no problem, I'll do it for you. I think for me at least I think (laughs)… I think for Chinese family, Chinese culture I can see, yeah, more… I don't know the other way but for Chinese people I guess it's easier to do this within the family." *Participant 14* |
| | | I mean if you suggest anything about my children, my wife or very close family if they die, if that is the only way I can save them I will have no hesitation to do it. But then if it's not my family member I would be thinking hang on a minute, why his own or her own family member not doing it. And also because I still have a responsibility to my own family" *Participant 8* |
| | Relationship hierarchy | Definitely close-knit family, probably extended family and probably close friends and I guess it kind of depends, you know, I probably would be more inclined to donate to a close family member than someone who've I probably met once or twice but if the need is there then, yeah, I'd consider it." *Participant 5* |
| | | "I guess within the family its easier because, let's say if my close family members got a problem, it's easier to make a decision but if its outside of friends or outside of family it will be difficult because it's difficult to understand how complicated the operation is or if it's safe or ….will I get some problems later on when I get older if I just have one kidney" *Participant 15* |
| | | "I think most people don't really think about giving a part of them to help others, especially other people who aren't of their own family. You'd have to be… I mean it is just incredible what some people… what sort of thought process and decision-making processes that they have to go through, what sort of transience that they had to reach, what sort of enlightenment that they had to achieve in order to reach that stage' *Participant 8* |
| | | But for a friend or somebody who's not that close, if you want to do it it's nice… it's really a kind, how you say it, it's a kind… you know, it's something that you are going to contribute to the community but not that close or similar to what happens in your family. *Participant 14* |
| | Matching | "if it's a family member or someone who's very close then it means that you can get like a closer match to the person who's receiving the kidney so there's less chance of rejection…if there'd be less chance of rejection then I think it would be a good thing to consider. Apart from that it shouldn't really matter who you donate to, whether they're Chinese or not." *Participant 5* |
| | | "I think it's very important because I think the Chinese people, because we were born in the same place and similar food, have the similar habit, so I think maybe condition of our health will be very similar so less risk of the rejection, if the Chinese people transplant to another Chinese people I think." *Participant 1* |
| | | "From my personal perspective I don't care if I need and I will not care that people with me is Chinese people or other foreign people for me because I will very appreciate if the people give the kidney to me. From my personal perspective I will not care." *Participant 1* |

*(Continued)*

**Table 2.** (Continued)

| Theme | Subtheme | Quote |
|---|---|---|
| Donor sacrifice | Negative impact on donors | "I think it's only a personal thing, is losing one kidney, your health will go downhill then you've got to balance or reassess or think about is it worth to do it." *Participant 8* |
| | | "one single kidney and basically your body needs both of them, that's why you're supplied with both. But if you are going to reduce it by half then something else is going to give."<br>"people may be afraid of facing the future with only one kidney, I mean when you hear of someone losing a kidney its already an illness and for you to donate a kidney is purposefully making yourself ill so that somebody else can live" *Participant 15* |
| | | "I think it's a really personal sacrifice to make and I think I wouldn't be upset if no-one that I knew would donate a kidney to me because it's kind of… it's a really big operation and then you've got long-term health implications so it's a really big sacrifice to make" *Participant 5* |
| | | "where people might hesitate is because anything that's related to organ is related to surgery, anything related to surgery is likely to be painful. So I don't think it's the donation part that's a problem for most people, I think for most people there is this fear of surgery, a risk of surgery and I think it's not just an ethnic community thing, I think there is like a high proportion of people who have fear of surgery or a fear of undergoing surgery so I think that's probably why, you know, as a clinician when you're mentioning organ donation, someone they might hesitate a little."<br>"So if there was an easy way to donate an organ without going through surgery then I think (laughs) most people would want to choose that direction but I think this whole… this fear of just undergoing the knife." *Participant 7* |
| | | "I suppose the hesitation is that in a sense that we kind of fear kind of… fear of going through the suffering to donate it." *Participant 2* |
| | Bodily integrity after death | "I think the major reason being because they think that the family themselves may not be able to accept the fact that your loved one is being cut up and take bits… organs and what have you taken away after death and then they do not want to upset the family, even though… yeah, I believe that is the major… well personally for me that would be one of the major issues I need to address if I was to go, you know, to donate any of my organs." *Participant 10* |
| | | "Spiritually I think they're kind of like, how should I put it, afraid of what would happen if they were cut up after passing, you know, what happens with the soul, will it remain as such or if you cut them up do they go to hell as opposed to going to heaven."<br>"when we're educated on the topic you need to tell them it's not something that you see in a movie, like you're being cut up into pieces and your kidney's going to be snatched away from you and then your body's going to be left in tatters or, you know, it's not… This is something that... I believe that a lot of people believe that's how it's going to happen." *Participant 6* |
| | | "a lot of Chinese people they don't accept this idea, like donating their organs. If you remember my parents or elderly they consider after you've passed away they would like to be buried in piece of land, like… they want to find somewhere…to bury your ancestor. So this is the culture as well. But nowadays I think young people they are more open-minded and if you pass away, you pass away, you know, no life and I think we're not, you know, do this we can contribute to other people, we contribute to the world. I think its something that more people starting to think that this is something that actually is acceptable." *Participant 14* |
| | | "it's the culture rather than religious belief I think because everyone says oh you've got to die whole, with your whole body, and because the cremation we want whole body probably for the next life, I don't know. It's changing.....well I think your generation have, because the Western cultures influence, they do understand more about organ donation and they change people's mind." *Participant 8* |
| | | "I believe it's fifty-fifty. Some of them have a religious belief, some of them just don't want to give out any body parts to anyone and they don't even want to lose a hair, but for me after I die I don't care, if any of my body parts can help people I would donate." *Participant 11* |

*(Continued)*

**Table 2.** (Continued)

| Theme | Subtheme | Quote |
|---|---|---|
| Patient as information gatekeeper | Culture of silence | "Most of the time we talk about something happy, we don't want to talk about a topic like this, too serious. Although we do have guest talk about organ donation but we only listen, after that we don't really discuss among ourselves." *Participant 16* |
| | | "So we don't want to kind of intrude upon other people's kind of areas which might make them feel uncomfortable." *Participant 2* |
| | | "So it was actually my friend's mum and I knew she'd been ill for a while but she hadn't really said what was wrong with her" *Participant 5* |
| | | "I guess again fifty-fifty. Some when they get older they don't want to chat about the subject, death or disease, however some other people may want to know more about it, get some more information and that's how I feel." *Participant 11* |
| | | "you're not asking to borrow money from them, you're asking part of their, you know, the body and you're asking for a perfectly healthy person to… It's a great sacrifice isn't it? It has to come from them and I don't feel comfortable asking, no not even to my children, no." *Participant 6* |
| | | "if someone of my family wanted to donate one to me I probably would say well hang on to that, well everyone has to die and I don't want to be selfish by taking other people's organs and make their life difficult. Obviously this is what I'm thinking now. I'd be reluctant to… I wouldn't even ask." *Participant 8* |
| | A perceived need for education and engagement | "So there's uncertainties about the future for oneself, so I guess until we get more information, more education, then I don't think there will be a lot of people who are willing to do this." <br> "I think if we want to promote this, want more people to recognise the importance of the kidney transplant or we need more people who want to donate kidneys, I think maybe more, how you say, more education in terms of like for example print out some leaflets or posters, put up on the supermarket board or hospital noticeboard to attract people's attention so people will read and think about it." *Participant 14* |
| | | "I think we need to be educated and well-informed of the whole process and I think that could change many people's opinions on it." *Participant 6* |
| | | "So I think the idea of having an opt-out system is good because it means that unless people specifically opt-out and it means that they would have seriously thought about that and made a decision to opt out for whatever reason it means that people who don't necessarily get round to doing it and who might well have intended for their organs to be donated but never actually made that wish," *Participant 5* |

## Importance of kinship: biological and social

All study participants emphasised that a close personal relationship between donor and recipient was important for living kidney donation (LKD), but participants didn't talk about a sense of kinship being important for donation after death.

**Familism.** When discussing willingness to partake in LKD, most participants afforded special status to family members, referencing the centrality of the family within Chinese culture. Willingness to donate appeared to be driven by affection for family members and a sense of duty. When asked about donating to people outside of the family (e.g., friends, colleagues) several participants described how this was the responsibility of the individual's own family members, thus applying their own sense of duty to others.

When asked about receiving a living-donor kidney transplant, many participants expressed concerns that family members would feel obliged to "make sacrifices for them". This seemed to reflect two ideas: the notion that individuals would feel duty-bound to donate (due to familial responsibility); and the notion that LKD has negative implications for the donor. The theme of donor sacrifice is discussed in more detail below. These beliefs made individuals reluctant to discuss kidney donation with close relatives, contributing to a 'culture of silence'.

> "you're not asking to borrow money from them, you're asking part of their, you know, the body and you're asking for a perfectly healthy person to… It's a great sacrifice isn't it? It has to come from them and I don't feel comfortable asking, no not even to my children, no." *Participant 6*

One participant was willing to accept an organ from their parents but justified this on the grounds that improvements to their personal health would leave them better placed to fulfil their filial duties. In other words, the

perceived risks of kidney donation were accepted because donation was seen as bringing additional benefits to the donor:

> "if my parents just had me, just had one child and they need me to accompany with them and to take them when they old so maybe I think they need me….I will like to accept their kidney because I know without me they cannot live here" *Participant 2*

**Relationship hierarchy.** For several participants, attitudes to LKD were influenced by a relationship hierarchy, with willingness to donate correlating with the emotional, social or biological 'closeness' to the proposed recipient. Some participants described a relationship gradient, with hypothetical willingness to donate gradually reducing as the closeness to the proposed recipient diminished. Other people described a relational/relationship threshold: a defined social distance above which, for them, the risks of donation outweighed the benefits. One participant felt people didn't typically consider donation outside the family and described individuals considering donating in this way as being 'enlightened'. This view suggested donation outside the family unit was not seen as a behavioural norm.

> "I think most people don't really think about giving a part of them to help others, especially other people who aren't of their own family. You'd have to be… I mean it is just incredible what some people… what sort of thought process and decision-making processes that they have to go through, what sort of transience that they had to reach, what sort of enlightenment that they had to achieve in order to reach that stage' *Participant 8*

**Matching.** Many participants described the importance of achieving a close 'match' to avoid rejection, although the meaning of matching seemed to vary among participants. Many believed close family members would be better matched; however, it was difficult to disentangle the relative importance of matching and the influence of familial duty in explaining the strong preference for within-family donation.

Some participants described shared ethnicity between donor and recipient as being important in terms of a higher likelihood of a favourable match, although many did not seem to have a preference about donating to or receiving a kidney from someone of the same ethnicity. For those who felt shared ethnicity was important, positive clinical outcomes seemed to be main motivator behind this belief, rather than the notion of kinship or shared responsibility amongst individuals of Chinese ancestry.

> "there'd be less chance of rejection then I think it would be a good thing to consider. Apart from that it shouldn't really matter who you donate to, whether they're Chinese or not." *Participant 5*

## Donor sacrifice

Most participants viewed donation, both in life and after death, as a personal sacrifice for the donor, and something that should be weighed carefully against potential benefits to the recipient. Participants did not appear to perceive additional benefits to the donor beyond improving the health of the recipient. Several participants mentioned how kidney transplants may fail or were unlikely 'to last forever' and this seemed to heighten the notion of donor sacrifice.

**Negative impact on donors.** Many participants expressed concerns about potential negative short- and long-term effects on living kidney donors. For example, several participants expressed the view that LKD negatively impacts the long-term health of a donor. Some ascribed significance to having "two kidneys for a reason", although one participant stated that "we can rely on one". One participant described how donating a kidney would constitute "purposefully making you ill" resulting in "long-term health complications". No participant stated explicitly that they were worried about the

increased risk of kidney failure, however reference to the importance of having two kidneys did seem to imply that participants were concerned about the impact of donation on kidney function. Many participants described concerns about physical pain or suffering in the immediate aftermath of major surgery. One participant described how this was not "just an ethnic community thing" but something relevant to individuals more broadly.

**Bodily integrity after death.** Many participants talked about the theoretical importance of keeping the body intact after death. Most viewed this as a barrier to deceased donation for other individuals in the UK Chinese community, but no participant described it as influencing their own personal views towards organ donation. Participants reported several different perceived reasons for people believing bodily integrity was important. Removing organs after death was described by some participants as constituting a form of mutilation, with the potential to cause distress for the surviving family. Other participants described the idea that the body not remaining whole could impact a donor's soul and transition through the afterlife. Several participants implied that organ donation may preclude burial and necessitate alternative funeral practices (e.g., cremation), which were described as less acceptable in Chinese culture. Participants felt concerns about violating bodily integrity were more prevalent amongst older generations and changing with the influence of "Western culture".

"it's the culture rather than religious belief I think because everyone says oh you've got to die whole, with your whole body, and because the cremation we want whole body probably for the next life, I don't know. It's changing.....well I think your generation have, because the Western cultures influence, they do understand more about organ donation and they change people's mind." *Participant 8*

However, the views of our individual participants were not in keeping with this pattern, as older participants and those born in China didn't share strong beliefs about bodily integrity. Discussions regarding bodily integrity focussed on donation after death and no participant discussed this as an important factor to consider with respect to LKD.

## Patient as information gatekeeper

**Culture of silence.** Several participants described how a reluctance to discuss ill health was embedded in Chinese culture, creating a culture of silence that naturally restricted discussions around kidney failure, transplantation, and organ donation.

"So we don't want to kind of intrude upon other people's kind of areas which might make them feel uncomfortable." *Participant 1*

Many described how they believed that the onus was on those with kidney disease to initiate discussions about donation. However, participants also suggested that they themselves would be reluctant to reach out to potential living donors if they were personally living with kidney disease, due to concerns about pressurising others to make a significant personal sacrifice. As with the theme of bodily integrity, one participant described how they believed that the culture of silence was changing among UK Chinese people due to the impact of 'Westernisation', but felt this cultural idea remained engrained in mainland China.

**A perceived need for education and engagement.** Several participants stated that greater access to information was needed to improve willingness to engage in organ donation. Some respondents described the importance of a small number of 'highly trusted' institutions in providing information, often directly referencing the UK National Health Service. Despite this perceived need for greater knowledge and awareness at a population level, no-one described an intention to improve their personal understanding of organ donation. In an acknowledgement of this pattern of behaviour, one participant described their enthusiasm for the 'opt-out' law in terms of its ability to widen access to transplantation in an

environment where individuals fail to seek information and/or formalise their views. The disconnect between participants expressed views and their own behaviours seemed to imply other factors were influencing their willingness to seek out information, although these were not explicitly described by participants. Hesitance to discuss issues relating to organ donation with family, and a broader culture of silence, may both have been relevant and are described in detail above. In addition, the emphasis on 'trust' implied participants were concerned about this risk of misinformation, and this could reduce individuals' willingness to conduct research independently.

## Discussion

To our knowledge, this is the first qualitative study to explore the views of UK Chinese individuals to deceased and LKD. Participants described several important concepts that influenced decision-making, and these may help explain the lower relative LKD rates among UK Chinese reported in our parallel UKRR analysis. A sense of familial duty and the perceived need for a close match between donor and recipient appeared to explain a low willingness to consider unrelated living donation. LKD was perceived as negatively impacting donors, associated with short- and long-term harms, leading to the prevalent notion that donation is a significant personal sacrifice. Combined with a broader 'culture of silence' among UK Chinese individuals with respect to ill health, this belief appeared to stymy open discussions about living and deceased donation/transplantation. We did not find an obvious explanation for the lower likelihood of pre-emptive listing and transplantation in UK Chinese men reported in our UKRR analysis. With respect to deceased organ donation, maintaining bodily integrity after death was considered by participants as a barrier to donation in Chinese society, although no individual reported that they personally held this belief.

Our participants voiced a strong willingness to donate to close family members, often citing their duty to family. In Chinese culture, this duty is associated with the Confucian notion of familism, describing how family has ontological priority over individual family members [13,14]. Although no participant explicitly referenced familism, our findings suggest this Confucian idea remains prominent in the UK Chinese community, as has been reported in other ethnically Chinese populations outside mainland China [15]. In a survey of 100 Chinese Americans, 83.6% strongly agreed that they would donate to close family members after death, falling to 45.9% for distant family members, and 28.7% for strangers [15]. A reduced willingness to consider donors outside the family unit may explain the reduced rates of living-donor kidney transplantation in UK Chinese individuals, particularly given the UK Chinese population are disproportionately young, having emigrated to the UK to work or study, meaning their family pool of potential donors may predominantly be living abroad [4,16].

Many of our participants discussed the belief that kidney donation violates bodily integrity. As a perceived barrier to organ donation, this has been widely reported in individuals of Chinese ethnicity – both in mainland China [17,18]) and outside of China [15,19,20] – and in other ethnic groups, including the UK White [21–23] and the UK Black population with Caribbean heritage [24]. Our participants reported being uncertain as to the basis of 'traditional' views, reflecting a complex interplay between several different religious/cultural beliefs, as has been described in other Chinese communities outside mainland China [19]. Relevant ideas in Chinese culture include the Confucian notion of filial piety (describing a need for children to return their bodies in the same condition as they received them out of respect for their parents); the Buddhist belief that the spirit can be injured by disturbing the body after death; the Taoist view that interference with the body can upset the balance of ch'I; and the belief that organ donation may prevent burial (which is preferred over cremation in traditional Chinese culture) [15,25]. Several participants drew a distinction between their personal views on bodily integrity and those that they perceived as being held by the wider Chinese community. Participants felt the importance of traditional beliefs was waning, particularly in the young. A similar acculturation process has been reported in individuals of Chinese heritage in the US, particularly with increasing time following emigration [26]. Despite perceived attitude changes, cultural/religious beliefs are still the most common reason for families of UK Chinese individuals to refuse consent for donation, accounting for 33.3% (n = 19/57) of refusals between 2015–2020 [27]. This is greater than the proportion in the UK White population (1%, n = 43/4425)) and in other UK minority ethnic groups (31.2%[n = 157/503] Asian;

19.2%[n = 57/297] Black). However small overall numbers in the UK minority ethnic groups, particularly the UK Chinese group, means caution is needed when interpreting proportional differences [27]. In our study, concerns about bodily integrity centred on deceased donation and no participant mentioned that it was a barrier to living donation.

Most participants shared the view that LKD negatively impacts donors and only one respondent described positive consequences for a donor from donation. No participant discussed the long-term risks of LKD in detail, nor quantified the risks, although many were concerned about short term physical suffering in the aftermath of surgery. The risks of LKD are low: peri-operative mortality is between 0.01% to 0.03%, the absolute 15-year incidence of kidney failure is less than 1% for most donors, and donation does not seem to impact long-term health-related quality of life [28–32]. A systematic review of 27 quantitative and 6 qualitative studies in East Asian individuals in the US did not identify donor health concerns as a barrier to organ donation, but this only included studies pre-2001, before the risks of LKD had been better estimated and widely reported in large national cohort studies published from 2013 onwards [29,33,34]. We hypothesise that our findings might represent a temporal change in our understanding of the risks of kidney donation. Concerns about the risks of donation could reduce willingness of UK Chinese individuals to donate, but also their willingness to accept a living-donor kidney transplant, because of a perceived duty to protect the health of other family members.

In this study, participants described a cultural tendency to avoid discussions about ill health. This corresponds with findings in other Chinese communities: in a US survey of 278 respondents, Chinese Americans indicated greater reluctance to discuss organ donation with family compared with White Americans [35]. In a separate survey of 514 Asian Americans, only 25% reported having had a previous family discussion regarding organ donation registration wishes and those registered as organ donors where much more likely to have had a discussion compared to non-registered participants (56% versus 17%), implying greater openness may be associated with increased willingness to donate [36]. This 'culture of silence' is well-described as a barrier to transplantation in other UK minority ethnic groups [37]. Qualitative analysis of free-text questionnaire responses in a UK-based study of 1240 kidney patients from Black, Asian and other UK minority ethnic groups, identified 'a culture of silence' as a key barrier to LKD [38]. Interview studies in Chinese Canadians have reported that discussing topics related to death/dying, such as organ donation, are viewed as evoking bad luck in Chinese culture [20]. This may relate to the Confucian notion of 'ch'I': a fear that acknowledging illness could bring their family shame due to its perceived link with improper behaviour [14]. Individuals with kidney disease may also be reluctant to discuss organ donation with their families to leave them free from the burden of care under the Confucian notion of benevolence or 'ren'.

Participants in our study suggested family members would likely feel duty-bound to help others in their family through organ donation. Participants indicated they would be reluctant to initiate conversations with family about donation if they were suffering from kidney disease, so as not to impose this duty on others, particularly given organ donation was seen as negatively impacting donors. A similar concern among potential organ recipients has been reported in a retrospective analysis of 1273 potential kidney recipients in South Korea, where worries about the health of the potential donor were the most common recipient-related reason for discontinuation of the LKD process [39]. The 'culture of silence' extended to conversations around deceased kidney donation, suggesting concerns about imposing a 'duty to donate' on family members was not the only reason limiting open discussions, reinforcing the notion that reticence is culturally engrained at a deeper level. Consistent with this finding, UK transplant data indicate that uncertainty about whether a patient would have agreed to donation was the second most common reason for families of UK-Chinese individuals to decline consent to deceased kidney donation [27]. By contrast, families of South Asian and Black donors were more likely to state that a relative had previously expressed a wish not to donate, suggesting a prior discussion had taken place [27].

There are some important themes identified in previous research amongst UK minority ethnic groups and in other ethnically Chinese populations that did not emerge in our study. Lack of knowledge has been reported as a barrier to transplantation in two separate systematic reviews among minority ethnic groups in the USA [34,37]. This was not identified in our study; however, may be explained by 9/16 participants having university-level education, which broadly reflects the high educational attainment amongst UK Chinese compared to other UK ethnic groups [16].

## Strengths and limitations

To our knowledge, this is the first qualitative study to investigate views towards transplantation and organ donation in the UK Chinese population. Participants were diverse in terms of age, gender, place of birth, year of emigration to the UK (if applicable) and education level. The vast majority of individuals in the UK Chinese population are born outside the UK, which was reflected in our sample [16]. This study was conducted in parallel with a quantitative study, and this mixed-methods approach allowed us to compare findings from both studies, to generate hypotheses to explain our quantitative findings, and highlight quantitative results that were not explained by findings in our qualitative study.

Our study does have some limitations. The UK Chinese population is a heterogenous group comprising individuals who self-identify as being of Chinese ethnicity, including individuals who are of Chinese heritage born in the UK, and individuals of Chinese heritage born outside the UK who have migrated to the UK. Most individuals (8/16) in our sample were born in Hong Kong; with 4/16 born in Mainland China. Although we did not identify differences in responses between individuals in the two groups, findings are unlikely to transfer to Chinese populations in other countries. Interviews were conducted over the telephone, which may have affected rapport between the participants and interviewer and prevented non-verbal cues from being interpreted. However, telephone interviews are recognised to yield rich data and may be better suited to collecting data on sensitive topics due to a perceived sense of anonymity [40]. Finally, this study was conducted in parallel to our UK registry analysis, and therefore specific quantitative findings could not be fully investigated in an explanatory sequential mixed-methods design.

## Conclusions

This qualitative interview study describes the views of UK Chinese individuals towards kidney donation and transplantation, providing insights into reduced rates of living-donor transplantation in the UK. Further research is required to investigate observational research findings not explained here, and to develop effective strategies to improve treatment access and outcomes for UK Chinese individuals with kidney disease.

## Supporting information

**S1 File. Topic guide for participant interviews.**
(DOCX)

**S2 File. COREQ checklist.**
(PDF)

## Acknowledgments

The authors would like to thank all the study participants for giving their time and sharing their experiences and views.

## Author contributions

**Conceptualization:** Katie Wong, Pippa K. Bailey.

**Data curation:** Katie Wong, Pippa K. Bailey.

**Formal analysis:** Matthew Beresford, Katie Wong, Pippa K. Bailey.

**Investigation:** Matthew Beresford, Katie Wong, Pippa K. Bailey.

**Methodology:** Katie Wong, Mohammed Al-Talib, Pippa K. Bailey.

**Supervision:** Pippa K. Bailey.

**Visualization:** Matthew Beresford.

 

**Writing – original draft:** Matthew Beresford, Mohammed Al-Talib.

**Writing – review & editing:** Matthew Beresford, Katie Wong, Mohammed Al-Talib, Pippa K. Bailey.

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
