## [Decision Letter · Decision Letter 0]

18 Dec 2024

Dear Dr. Beresford

Thank you for submitting your manuscript to PLOS ONE. After careful consideration, we feel that it has merit but does not fully meet PLOS ONE’s publication criteria as it currently stands. Therefore, we invite you to submit a revised version of the manuscript that addresses the points raised during the review process.

Please submit your revised manuscript by Feb 01 2025 11:59PM.  If you will need more time than this to complete your revisions, please reply to this message or contact the journal office at plosone@plos.org . A rebuttal letter that responds to each point raised by the academic editor and reviewer(s). You should upload this letter as a separate file labeled 'Response to Reviewers'.A marked-up copy of your manuscript that highlights changes made to the original version. You should upload this as a separate file labeled 'Revised Manuscript with Track Changes'.An unmarked version of your revised paper without tracked changes. You should upload this as a separate file labeled 'Manuscript'.

We look forward to receiving your revised manuscript.

Kind regards,

Ala Ali

Academic Editor

PLOS ONE

Additional Editor Comments:

Thanks to the authors for their efforts.

My main concerns are:

1. How the term UK-Chineese affected the eligebility for the study

2. It would be intersting to have some data of other UK population, if possible to compare with (Indian or African). Authors may add some data especially if those works used simillar methodology.

Reviewers' comments:

Reviewer's Responses to Questions

**Comments to the Author**

1. Is the manuscript technically sound, and do the data support the conclusions?

Reviewer #1: Yes

Reviewer #2: Partly

2. Has the statistical analysis been performed appropriately and rigorously?

Reviewer #1: Yes

Reviewer #2: N/A

3. Have the authors made all data underlying the findings in their manuscript fully available?

Reviewer #1: No

Reviewer #2: Yes

4. Is the manuscript presented in an intelligible fashion and written in standard English?

Reviewer #1: Yes

Reviewer #2: Yes

Reviewer #1: This is an important study and I enjoyed reading the paper. I have a few comments

Major

Based on the aims, the team wanted “to investigate and understand the attitudes of UK Chinese individuals towards kidney donation and transplantation.” Four concepts are being explored: being a living kidney donor, being a deceased donor, accepting a kidney from a deceased donor, accepting a kidney from a living donor. These are 4 different research questions and ethnic barriers and perceptions ought to be explored separately to truly understand the participants perceptions. For example, sense of familial duty and the need for a close immunological match may be of relevance considering being a living kidney donor and accepting a living donor, respectively. Some segregation of these concepts in the analysis will likely generate unique themes and truly allow an in depth understanding of barriers to donation and transplantation in this community, AND increase the impact of this work.

The use of the word “UK Chinese” or UK Chinese individuals/people/men/women” seems a bit out of ordinary and perhaps may be politically incorrect especially given the fact that only 4/16 were born in Mainland China. May be acceptable. I am unsure as I do not belong to this community. I would confirm with a native individual how do would like to be addressed in scientific literature. Also, I would read the paper in JAMA that made recommendations on how to report race and ethnicity

The team incorrectly uses the terminology for gender. It should be woman and man, not male and female

Other

The discussion is also quite lengthy and I instead would suggest making it more succinct, and perhaps consider expanding on the results.

This is also an educated group of participants, most of whom spoke English, hence I would suggest addressing a potential sampling/representative bias

Reviewer #2: The authors describe a qualitative study as a component of a mixed methods approach to understand perspectives of UK-Chinese who need a kidney transplant. The manuscript makes important contributions to understanding cultural influence on access to transplant. However, additional clarifications to interpret the themes could be made to further strengthen the manuscript.

Comments

The authors could clarify the methods to analyze data from translated interviews. The methods mention KW verified translations. After this step, was participant commentary no longer used and transcriptions and coding used only translator text (e.g. only English was transcribed)?

The topic guide includes a question about personal relationships impacted by kidney disease, but this is not captured in the results from themes or questionnaires. It is unclear if this question was coded. Some quotes suggest detailed knowledge of kidney disease and other quotes suggest significant misunderstandings. Information about the sample’s familiarity with the topic would be relevant.

The sample quotes provide a rich qualitative foundation, but some themes and subthemes seem better supported than others. Negative impacts on donor health theme is complicated by potential exaggerated perceptions of donor impacts, and this may be a relevant distinction. In addition, quotes related to fear may not be specifically related to donor health as much as the acute experience. The description of this theme also reinforces the concept of abstract suffering, suggesting a theme label that is broader than “donor health” may be appropriate.

The Passivity theme is most difficult to interpret. The description implies that since support for transplantation was only voiced in terms of what others could do and not what they could do, that participants were passive. This is somewhat contradicted by quotes that participants would certainly donate to family if asked. A potential limitation of this theme is that the topic guide does not seem to have questions that would reasonably elicit a meaningful response for the extent that the participant was passive. A question asks where information could be provided. If Quote #3 (e.g. discussion of leaflets in supermarkets) was in response to this type of question, it is potentially misleading to assign a meaning related to willingness to personally act or not act.

There is a typo on p 26, line 285.

The discussion includes a valuable review of related literature and how current findings compare to the body of research. The authors cite studies of Chinese Canadians; however, several more recent studies of South Asian and Chinese Canadians populations could provide additional insight.

The discussion touches on the potential tension between Familism and Culture of Silence, but this seems under explored and potentially of interest. This seemingly describes a perception that donation by others to family is a sacrifice more than a duty but a donation by oneself to family is more a duty and less a sacrifice. While this may be a poor articulation as a western reader, it would be a potential issue to address further.

**Do you want your identity to be public for this peer review?** For information about this choice, including consent withdrawal, please see our Privacy Policy

Reviewer #1: No

Reviewer #2: No

---

## [Author Response · Author response to Decision Letter 1]

29 Apr 2025

Please find attached 'response to reviewer comments' document which provides comprehensive responses to each comment raised.

---

## [Decision Letter · Decision Letter 1]

19 May 2025

The views of UK-Chinese individuals towards living and deceased-donor kidney transplantation: A qualitative interview study

PONE-D-24-49514R1

Dear Dr. Beresford, 

We’re pleased to inform you that your manuscript has been judged scientifically suitable for publication and will be formally accepted for publication once it meets all outstanding technical requirements.

Kind regards,

Ala Ali

Academic Editor

PLOS ONE

Additional Editor Comments (optional):

Reviewers' comments:

Reviewer's Responses to Questions

**Comments to the Author**

Reviewer #1: All comments have been addressed

Reviewer #2: (No Response)

2. Is the manuscript technically sound, and do the data support the conclusions?

Reviewer #1: Yes

Reviewer #2: Yes

3. Has the statistical analysis been performed appropriately and rigorously?

Reviewer #1: N/A

Reviewer #2: N/A

4. Have the authors made all data underlying the findings in their manuscript fully available?

Reviewer #1: (No Response)

Reviewer #2: (No Response)

5. Is the manuscript presented in an intelligible fashion and written in standard English?

Reviewer #1: Yes

Reviewer #2: Yes

Reviewer #1: I could not see the response to reviewers documents but based on the track document, it seems most of my comments addressed

Good luck

Reviewer #2: The authors provided significant revisions and improvements in response to the original review. Major concerns have been addressed.

A previous comment to consider more recent studies with Chinese Canadians did not include specifics to allow authors to identify potential publications. A few examples include: Alvin Li et al 2015, Pol et al 2024

**Do you want your identity to be public for this peer review?** For information about this choice, including consent withdrawal, please see our Privacy Policy

Reviewer #1: No

Reviewer #2: No

---

## [Editor Report · Acceptance letter]

PONE-D-24-49514R1

PLOS ONE

Dear Dr. Beresford,

I'm pleased to inform you that your manuscript has been deemed suitable for publication in PLOS ONE. Congratulations! Your manuscript is now being handed over to our production team.

Kind regards,

on behalf of

Dr. Ala Ali

Academic Editor

PLOS ONE